# Characteristics and Distribution of Radiologists in Saudi Arabia: A Cross-Sectional Study Based on National Data

**DOI:** 10.3390/healthcare13202651

**Published:** 2025-10-21

**Authors:** Jaber Hussain Alsalah

**Affiliations:** 1Department of Radiological Sciences, Faculty of Applied Medical Sciences, King Abdulaziz University, Jeddah 21589, Saudi Arabia; jhalyami@kau.edu.sa; 2Smart Medical Imaging Research Group, King Abdulaziz University, Jeddah 21589, Saudi Arabia; 3King Fahd Medical Research Center, King Abdulaziz University, Jeddah 21589, Saudi Arabia

**Keywords:** radiology, workforce, characteristics and distribution, health profession, specialization

## Abstract

**Background:** In healthcare institutions, radiologists play an essential role in patients’ care, enabling them to begin treatment and start their recoveries. However, data on the characteristics and distribution of the radiology workforce in Saudi Arabia are limited. Therefore, this study aimed to conduct a comprehensive analysis of the radiology workforce in SA based on national data and identify key distributional and specialty trends relevant to workforce planning and radiology service delivery. **Methods:** The following data were obtained from the Saudi Commission for Health Specialties (SCFHS) Registry: total number of registered radiologists, age, subspecialty, professional classification, place of qualification, and geographical location. Descriptive statistics were used for data analysis. Additionally, the findings were compared with those of published international benchmarks. **Results:** There were 5150 radiologists registered with SCFHS in SA, which corresponded to 147 radiologists per 1,000,000 inhabitants. The mean age was 40.8 years (standard deviation [SD] 9.8), with 60% of them being aged 30–44 years. Most of the radiologists specialised in general diagnostic radiology (83.7%), with few of them specialising in interventional radiology (1.8%), paediatric radiology (1.1%), and breast imaging (0.9%). The workforce mainly comprised consultants (35.0%), followed by registrars (29.7%) and senior registrars (22.7%). Two-thirds (65.0%) of the radiologists had obtained their qualifications abroad. More than half of the radiologists resided in three provinces: Riyadh (29%), Mecca (23%), and the Eastern Region (15%), while several provinces had fewer than 2% of the available workforce. **Conclusions:** The radiology workforce in SA is relatively young and has a higher density than the average in the European Union. Further, most of the radiologists are professionally classified as consultants or registrars. However, there is a clear imbalance in their geographic distribution, which is consistent with the population sizes of the respective cities. Targeted training expansion and reduced reliance on foreign-trained professionals are warranted to meet future service demands in line with the Vision 2030 objectives.

## 1. Background

Radiology plays a crucial role in contemporary healthcare by supporting early diagnosis, treatment planning, and follow-up across medical disciplines [1,2,3,4,5]. The size, distribution, and specialties of the radiology workforce directly influence access to radiology services, as well as timely and high-quality imaging, with implications for patient outcomes, treatment management, and performance efficiency of healthcare systems [6,7]. There has been a recent increase in the demand for radiological services due to demographic growth, epidemiological shifts towards chronic diseases, and the expansion of image-guided therapies [8,9,10,11].

There are substantial among-country variations in the availability of radiologists. For example, the mean density in the European Union (EU) is 127 radiologists/million (M) inhabitants, ranging from <100/M in some Eastern European countries to >250/M in Sweden and Greece [12,13]. These figures are relatively lower in low- and middle-income countries (LMICs); specifically, Ghana has a mean density of 1.7/M [14], while it is <1/M in several other sub-Saharan countries [15]. Workforce shortages are frequently compounded by misdistribution, with radiologists being relatively concentrated in large urban centres [13,14,15,16,17,18]. There was variability in radiology workforce supply across Gulf Cooperation Council (GCC) countries. For instance, radiologist density is reported at 90 per million populations in the United Arab Emirates and 110 per million in Qatar [19,20].

Saudi Arabia (SA) is currently undergoing unique and transformative economic and social reforms. Specifically, under the Health Sector Transformation Program of Vision 2030, there is an objective to enhance workforce capacity and quality of care within healthcare services. This reform has driven rapid healthcare expansion under Vision 2030, making a comprehensive assessment of the radiology workforce essential to guide planning and address subspecialty and geographic disparities [17]. Additionally, there are undergoing transformations in the healthcare system of SA to make it more comprehensive, effective, and integrated. This enhanced system prioritises innovation, financial sustainability, and disease prevention while improving access to healthcare. Furthermore, there is focus on expanding e-health services and digital solutions, improving quality of care, and adhering to international standards. Accordingly, there have been extensive investments in healthcare infrastructure and workforce development in SA [17,18,19,20,21,22]. Previous national studies have described the characteristics and distributions of healthcare professionals, including emergency medical physicians and respiratory therapists [23,24]. However, there remain limited national-level data regarding the geographic distribution, subspecialty representation, and professional ranking of radiologists.

As aforementioned, radiologists tend to be concentrated in urban regions, which leads to disparities in access to radiology services among rural populations [25,26]. This is compounded by the fact that advanced imaging equipment and sub-specialist expertise are often unavailable outside major cities. Sub-specialty composition is another determinant of service capacity [26]. For example, studies have reported shortages in radiologists specializing in interventional radiology, paediatric radiology, and breast imaging [27,28,29]. Addressing such gaps is relevant in SA, where service expansion in oncology, women’s health, and paediatric care is anticipated in the coming years [25,29,30].

Despite the importance of these issues, there is a lack of comprehensive national-level studies describing the radiology workforce in Saudi Arabia. To the best of our knowledge, no previous study has provided a comprehensive, up-to-date national overview of the radiology workforce in SA, including workforce density, demographic characteristics, academic backgrounds, subspecialty representation, and geographic distribution.

Accordingly, this study aimed to conduct a comprehensive analysis of the radiology workforce in SA based on national data as of August 2025, and identify key distributional and specialty trends relevant to workforce planning and radiology service delivery.

## 2. Methods

### 2.1. Ethics Approval

This study was approved by the Institutional Review Board of King Abdulaziz University Hospital (Reference No 170-25; date of approval: 23 April 2025) in Jeddah, SA.

### 2.2. Study Design

In this cross-sectional analysis, current data were sourced from the Saudi Commission for Health Specialties (SCFHS) as of 3 August 2025. The SCFHS oversees the classification and registration of all healthcare providers in SA, which is mandatory. Accordingly, the SCFHS is among the most reliable sources for acquiring accurate data in the radiology workforce.

Inclusion criteria were all practicing radiologists registered with the SCFHS at the time of data extraction. Exclusion criteria were duplicate entries, retired or non-practicing radiologists, and records with incomplete registration that precluded reliable identification. Cases with missing or “unknown” values for specific variables were retained in the overall dataset but excluded from subgroup analyses requiring those variables.

In April 2025, a formal request was made to the SCFHS data department. This study collected data regarding demographic and professional characteristics, including specialization, professional rank, region and province of employment, and educational background. Radiologists were categorized into five major work regions (Central, Eastern, Western, Northern, and Southern), which covered 13 administrative provinces.

Specializations were categorized as diagnostic radiology and the following subspecialties: body imaging, diagnostic neuroradiology, musculoskeletal radiology, interventional radiology, paediatric radiology, vascular and interventional radiology, cardiothoracic imaging, breast imaging, nuclear medicine, interventional neuroradiology, women’s imaging, cardiac imaging, molecular imaging, paediatric neuroradiology, thoracic imaging, emergency radiology, paediatric interventional radiology, nuclear medicine and molecular imaging, and interventional oncology.

Employment positions were classified into six professional ranks: consultants, senior registrars, registrars, resident-in-training, residents, and fellows. Furthermore, this study collected information regarding the type of academic institution from which each radiologist obtained their degree, which was categorized as local governmental colleges, local private colleges, and international institutions outside SA (referred to as “abroad”).

### 2.3. Statistical Analysis

All statistical analyses were performed using SPSS v26 software. Descriptive statistics were used to summarise the distribution of radiologists across regions, ranks, and specialties. Frequencies and percentages are reported to present the overall patterns and identify regional or specialty-related disparities. The associations between categorical variables were assessed using the chi-square test. All data are presented in tabular and graphical formats.

## 3. Results

### 3.1. Workforce

As of August 2025, there were 5150 radiologists registered with the SCFHS, corresponding to a density of 147/M, as shown in Figure 1.

As shown in Figure 1, there were 13 cities with a radiologist density higher than the EU average (127/M). Further, the radiologist density in SA ranged from 101/M in Jazan to 173.1/M in Riyadh. In Figure 1, the regions were classified into those with a radiologist density of 100–124.9/M (Jazan, Tabuk, Medina, and Asir), 125–149.9/M (Jouf, Baha, Hail, Eastern Region, and Qasim), and 150–174.9/M (Najran, Mecca, Northern Borders, and Riyadh).

### 3.2. Age Profile of Radiologists

The mean age of the radiologists was 40.8 years (SD = 9.8). Most of the radiologists were in the age group of 35–39 years (n = 1246; 24.2%), followed by 40–44 years (n = 964; 18.7%), and 30–34 years (n = 882; 17.1%). These three age groups accounted for 60% of the workforce (Figure 2). Radiologists aged 25–29 years and 45–49 years comprised 10.7% (n = 553) and 11.3% (n = 584) of the workforce, respectively. Relatively few radiologists were aged 50–54 years (n = 390; 7.6%), 55–59 years (n = 271; 5.3%), and ≥60 years (n = 260; 5.0%), which indicated a predominantly young workforce.

### 3.3. Professional Classification of Radiologists (Rank)

Figure 3 showed distribution of radiologist according to rankings. Consultants represented the largest group (n = 1801; 35.0%), followed by registrars (n = 1527; 29.7%). One individual (<0.1%) was registered as a general practitioner with radiology practice (Figure 3).

Regarding regional distribution, in all regions, the majority of radiologists were consultants, registrars, and senior registrars. As shown in Table 1, consultants were most prevalent in the Central (39.9%) and Eastern (39.1%) regions; contrastingly, there were least common in the Southern region (26.4%). Registrars were most prevalent in the Southern (37.8%) and Northern regions, with the lowest proportion being observed in the Central region (23.8%).

As shown in Table 1, these patterns reflect notable regional variation in workforce rank distribution. Senior registrars had the highest prevalence in the Southern region (26.8%). Regarding the employment sector, most residents-in-training (93.7%), consultants, and senior registrars (67.8%) worked in the public sector, while most of the residents (52.5%) and registrars (56.8%) worked in the private sector.

### 3.4. Specialties Distribution

Table 2 describes the specialisations of radiologists. The most common speciality was diagnostic radiology (n = 4313; 83.7%), followed by body imaging (n = 164; 3.2%), diagnostic neuroradiology (n = 126; 2.4%), musculoskeletal radiology (n = 94; 1.8%), and interventional radiology (n = 94; 1.8%). Other subspecialties each represented ≤1.1% of the workforce, including paediatric radiology (1.1%), vascular and interventional radiology (1.1%), cardiothoracic imaging (0.9%), breast imaging (0.9%), and nuclear medicine (0.6%). Rare subspecialties, including molecular imaging, paediatric neuroradiology, thoracic imaging, and emergency radiology, each comprised ≤0.2% of the workforce. There were regional differences in the proportion of radiologists specializing in diagnostic radiology (96.4% and 76.8% in the Northern and Central regions, respectively; 98.3% and 75.8% in the Northern Border area and Riyadh, respectively; Appendix A Table A1).

### 3.5. Education/Training

Approximately two-thirds (n = 3485, 67.7%) of the radiologists had studied abroad; further, 1518 (29.4%) and 47 (0.9%) radiologists had received education/training from public and private institutions in SA, respectively. Figure 4 shows the distribution of radiologists according to the institution of education and training.

### 3.6. Geographic Distribution

Regarding the geographic distribution of radiologists, the Central and Western regions accounted for 61.7% (n = 3175) of the workforce (Figure 5). Moreover, three large population provinces, including Riyadh (n = 1487; 29%), Mecca (n = 1200; 23%), and the Eastern Region (n = 762; 15%), accounted for 67% of the workforce.

Contrastingly, less populated provinces such as Al Jawf (n = 76; 1%), Al Baha (n = 44; 1%), and the Northern Borders (n = 60; 1%) each accounted for ≤2% of the workforce. The Southern region, which included Asir (5%), Jazan (3%), and Najran (2%), accounted for 11% of the workforce. The Northern region accounted for 6% of the workforce, mainly in Hail and Tabuk. Notably, the work location of 7% (n = 350) of the radiologists was unknown (Table 3).

Within the provinces, radiologists were concentrated in major urban centres. In Riyadh Province, 91.3% (n = 1357) of the radiologists were based in Riyadh City. In Mecca Province, Jeddah accounted for 59.0% (n = 708) of radiologists, followed by Mecca city (23.3%) and Taif (10.8%).

### 3.7. Professional Rank and Work Region

There was a significant association between professional rank and work region (χ^2^ test, *p* < 0.001). Consultants were most concentrated in the Central (39.9%) and Eastern (39.1%) regions, while their representation was lowest in the Southern region (26.4%). Registrars were disproportionately represented in the Southern (37.8%) and Northern (36.8%) regions, contrasting with the Central region (23.8%). Senior registrars showed a relatively balanced distribution, though they were proportionally highest in the Southern region (26.8%). The association between professional rank and work regions is statistically significant (*p*-value < 0.001), determined via the chi-square test. The detailed distribution is presented in Table 4.

## 4. Discussion

This study describes the profile of the radiologist workforce in SA, including workforce density, demographics, subspecialty distribution, professional classification, educational background, and geographic distribution. The radiologist density in SA (147/M) is higher than that in EU (127/M) [13,14] as well as many LMICs, including Ghana (1.7/M) [15] and Malawi (<0.5/M) [16]. Among other countries in the Gulf Cooperation Council, the radiologist density in SA exceeds that in the United Arab Emirates (90/M) and Qatar (110/M) [30,31]. This finding suggests that SA maintains a comparatively greater radiologist supply than some neighbouring high-income Gulf countries due to differences in population size and training capacity [19,20].

Contrastingly, the radiologist density in SA is lower than that in some European countries such as Norway (240/M) and Switzerland (230/M) [13], which demonstrates the potential for greater workforce expansion. Radiologist densities above 200/M have been associated with shorter imaging wait times and broader subspecialty coverage [14,31,32,33].

This study revealed an uneven distribution of radiologists across the country. Specifically, more than half of all radiologists were located in three provinces (Riyadh, Mecca, and Eastern Province), which can be attributed to their population sizes. Contrastingly, the Northern and Southern provinces accounted for ≤2% of the radiologist workforce. Similar distributions of radiologists have been observed in Europe [13] and sub-Saharan Africa [14,15], where the distribution of radiologists is associated with the location of training centres, specialist hospitals, and advanced imaging equipment.

Disparities in radiologist distribution within rural regions may limit access to diagnostic and interventional imaging, which contributes to delayed diagnosis as well as increased reliance on referrals to urban centres [26,27]. However, comparable urban/rural distributions have been described in Canada and Australia [33,34], which can be attributed to successful mitigation strategies, including rural practice incentives, decentralised fellowship rotations, and targeted equipment allocation [35]. Furthermore, urban–rural disparities can be addressed through teleradiology, which has been widely used to meet subspecialty coverage needs, balance workloads, and mitigate understaffing [36,37,38,39,40,41,42]. Experiences from understaffed regions highlight the role of teleradiology in maintaining continuity of care and facilitating access to advanced imaging interpretations across geographic boundaries [42]. Notably, teleradiology also improves the reporting time; enhances the efficiency of resource utilization; improves patient satisfaction in radiology consultations; and improves access to specialized care, cost savings, and collaboration [35,36,37,38,39,40,41,42].

In Saudi Arabia, workforce disparities may be further explained by the concentration of radiologists and advanced imaging facilities in major urban centres, the influence of population size, and limited incentives for rural practice [11]. Subspecialty shortages are of particular concern, especially in interventional radiology, paediatric radiology, and breast imaging, as they directly affect service delivery in priority areas such as oncology, women’s health, and paediatric care [3,43]. Addressing these disparities will require targeted workforce policies under the Vision 2030 health transformation program, including expanding subspecialty training, enhancing rural recruitment incentives, and strengthening national teleradiology infrastructure to ensure equitable service delivery [17].

In Saudi Arabia, the Ministry of Health (MOH) launched a national teleradiology platform in 2018 to support rural hospitals and primary healthcare, with early evidence indicating improved access and workflow efficiency [17]. Recent national evaluation shows that the platform has achieved broad regional adoption, with more than 90% of users reporting satisfaction and 92% recognizing its transformative impact on healthcare delivery [43]. The study also highlights substantial participation from northern and eastern regions, reflecting wide geographic coverage and increasing service utilization across the Kingdom [43]. Nonetheless, implementation is challenged by variations in digital infrastructure across regions, the need for robust medical data security frameworks, and limited integration of teleradiology into standardized workflows [43]. Importantly, such platforms can also be used as educational tools to leverage scarce expertise while supporting radiologists’ capacity building [43,44,45].

The radiologist workforce in SA is predominantly young, with 60% of them being aged 30–44 years. Contrastingly, nearly half of the radiologists in the EU are aged >51 years [11]. The relatively young age profile of radiologists in SA supports long-term workforce sustainability and the health transformation program. However, workforce youthfulness may represent challenges, particularly due to the limited number of senior radiologists available to provide mentorship and advance the large cohort of registrars and senior registrars to consultancy level. Therefore, it is important to ensure continued professional development, mentoring and structured career progression in order to ensure that healthcare systems have an appropriate radiology workforce for providing effective and efficient services [13,43].

In SA, most of the radiologists (83.7%) were specialised in general diagnostic radiology, with relatively few specialising in interventional radiology (1.8%), paediatric radiology (1.1%), and breast imaging (0.9%), which is consistent with international trends [28,29,30,31]. For example, interventional radiologists account for only 5% of the workforce in the United Kingdom [46,47], while paediatric radiology is almost entirely absent in Ghana [15]. These disparities limit the availability of advanced interventional procedures, paediatric imaging services, and breast cancer screening, especially outside major centres. Targeted fellowship expansion, international training partnerships, and early career exposure to subspecialties can help further mitigate these disparities [48].

Educational and training backgrounds also reflect the evolving nature of radiologists in SA. In this study, two-thirds of radiologists received education and training abroad. Although this may provide diverse training experience, it may also reflect a historical reliance on foreign-trained professionals [49] and vulnerability to international workforce mobility [50]. However, the relative youthfulness of the current workforce, coupled with a limited number of senior mentors, may hinder subspecialty development and affect the quality of postgraduate training. Increasing the domestic capacity in education and postgraduate training is essential for sub-specialty development. Additionally, it is important to consider patterns in other nations with rapidly expanding imaging services [13,15]. Strengthening local postgraduate and fellowship programs through partnerships and investment is crucial for reducing external dependency and retaining talent within the national system [49,51].

EU countries that are highly dependent on internationally trained specialists have faced recruitment challenges during periods of global shortage [13]. Reliance on foreign-trained radiologists carries important policy implications, particularly the need to expand local training programs, ensure standardized competencies, and strengthen national capacity for long-term workforce sustainability. Accordingly, expanding domestic residency and fellowship capacity, which is consistent with the SCFHS standards [1] and European Continuing Medical Education frameworks [52], is essential for long-term self-sufficiency. Additionally, incorporating telemedicine competencies into training can help prepare the workforce for the hybrid service models that are being increasingly employed to address subspecialty and geographic gaps [36,37,51,53].

This study has several limitations, including missing data and reliance on self-reported subspecialty information and its cross-sectional design, which limits causal inference and temporal analysis. The dataset did not specify the countries or regions where radiologists obtained their overseas qualifications. Future research should explore the origin nations of foreign-trained radiologists to better evaluate workforce mobility, training quality, and the impact on national capacity building. Future research should incorporate clinical workload indicators such as the number and types of imaging examinations performed, alongside sectoral distribution between public and private facilities to better assess service capacity. In parallel, the availability and distribution of imaging modalities should be evaluated, as they are critical determinants of workforce efficiency and service delivery.

## 5. Conclusions

The radiology workforce in SA is relatively young and has a higher density than the international average. Urban–rural disparities and slight shortages in some subspecialties indicate the need for further targeted workforce planning. Strategic interventions under Vision 2030 should focus on subspecialty development and expansion of domestic training to ensure nationwide access to high-quality imaging services. The current findings offer valuable insights into the composition of the radiology workforce, shedding light on the ongoing challenges and opportunities for development within this essential medical field.

## Figures and Tables

**Figure 1 healthcare-13-02651-f001:**
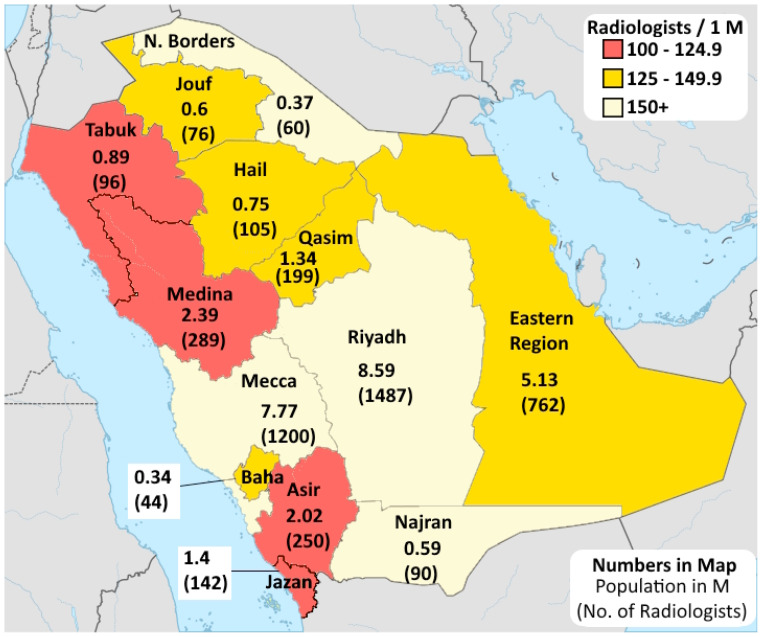
The geographical distribution of radiologists across SA.

**Figure 2 healthcare-13-02651-f002:**
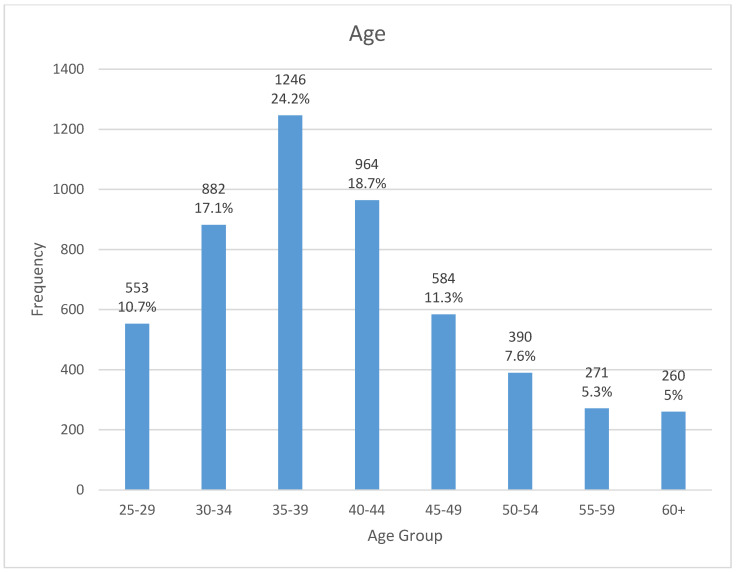
The age profiles of radiologists in SA.

**Figure 3 healthcare-13-02651-f003:**
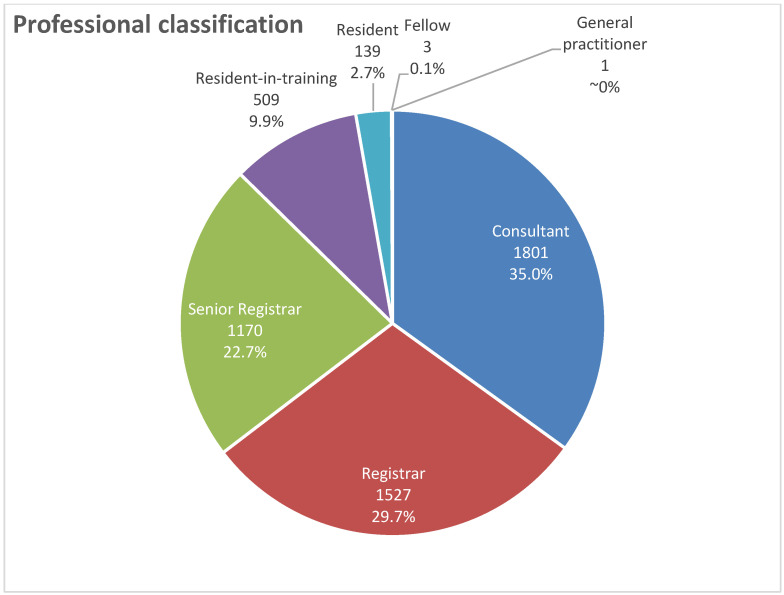
The distribution of radiologist according to rankings.

**Figure 4 healthcare-13-02651-f004:**
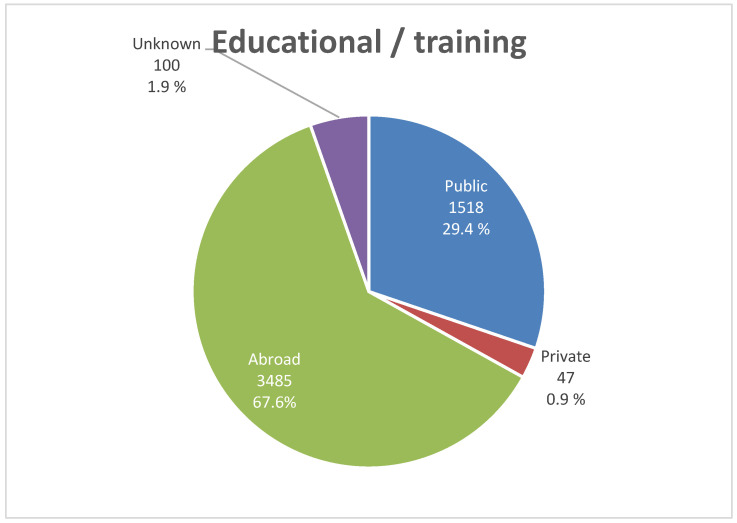
The distribution of radiologist according to education/training.

**Figure 5 healthcare-13-02651-f005:**
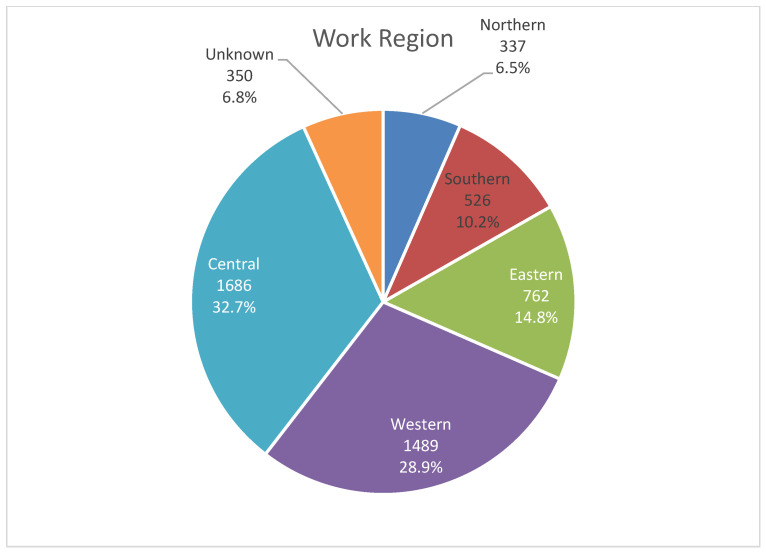
The distribution of radiologist according to work region.

**Table 1 healthcare-13-02651-t001:** Distribution of professional classifications according to the work regions in Saudi Arabia.

Rank	Statistic	Work Region					
		Northern	Southern	Eastern	Western	Central	Unknown
Consultant	Count	109	139	298	517	672	66
	% within work region	32.3%	26.4%	39.1%	34.7%	39.9%	18.9%
Fellow	Count	0	0	0	1	2	0
	% within work region	0.0%	0.0%	0.0%	0.1%	0.1%	0.0%
Senior registrar	Count	73	141	147	284	382	143
	% within work region	21.7%	26.8%	19.3%	19.1%	22.7%	40.9%
General practitioner	Count	0	0	0	1	0	0
	% within work region	0.0%	0.0%	0.0%	0.1%	0.0%	0.0%
Resident	Count	9	11	13	26	55	25
	% within work region	2.7%	2.1%	1.7%	1.7%	3.3%	7.1%
Resident-in-training	Count	22	36	50	200	173	28
	% within work region	6.5%	6.8%	6.6%	13.4%	10.3%	8.0%
Registrar	Count	124	199	254	460	402	88
	% within work region	36.8%	37.8%	33.3%	30.9%	23.8%	25.1%

**Table 2 healthcare-13-02651-t002:** Radiologist specifications.

Specialization	Number	%
Diagnostic radiology	4313	83.7
Body Imaging	164	3.2
Diagnostic Neuroradiology	126	2.4
Musculoskeletal Radiology	94	1.8
Interventional Radiology	94	1.8
Paediatric Radiology	59	1.1
Vascular and Interventional Radiology	56	1.1
Cardiothoracic Imaging	45	0.9
Breast Imaging	44	0.9
Nuclear Medicine	31	0.6
Interventional Neuroradiology	26	0.5
Women Imaging	25	0.5
Cardiac Imaging	10	0.2
Molecular Imaging	10	0.2
Paediatric Neuroradiology	10	0.2
Thoracic Imaging	10	0.2
Emergency Radiology	7	0.1
Paediatric Intervention Radiology	7	0.1
Nuclear Medicine & Molecular Imaging	6	0.1
Interventional Oncology	3	0.1
Other specialties	10	0.2
Total	5150	100.0

**Table 3 healthcare-13-02651-t003:** Work regions and provinces.

Work Region	Work Province	Number	Percentage
Northern	Hail	105	2%
	Jawf	76	1%
	Northern Border	60	1%
	Tabuk	96	2%
Southern	Asir	250	5%
	Baha	44	1%
	Jazan	142	3%
	Najran	90	2%
Eastern	Eastern Region	762	15%
Western	Mecca	1200	23%
	Medina	289	6%
Central	Qasim	199	4%
	Riyadh	1487	29%
Unknown	unknown	350	7%
Total		5150	100

**Table 4 healthcare-13-02651-t004:** Professional rank of radiologists and their working regions.

Rank	Work Region					
	Northern	Southern	Eastern	Western	Central	Unknown
Consultant	109	139	298	517	672	66
	32.3%	26.4%	39.1%	34.7%	39.9%	18.9%
Fellow	0	0	0	1	2	0
	0.0%	0.0%	0.0%	0.1%	0.1%	0.0%
Senior registrar	73	141	147	284	382	143
	21.7%	26.8%	19.3%	19.1%	22.7%	40.9%
Resident	9	11	13	26	55	25
	2.7%	2.1%	1.7%	1.7%	3.3%	7.1%
Resident-in-training	22	36	50	200	173	28
	6.5%	6.8%	6.6%	13.4%	10.3%	8.0%
Registrar	124	199	254	460	402	88
	36.8%	37.8%	33.3%	30.9%	23.8%	25.1%

## Data Availability

The datasets generated during and/or analyzed during the current study are available from the corresponding author on reasonable request due to privacy or ethical restrictions.

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
