# Peer review of "Characteristics and Distribution of Radiologists in Saudi Arabia: A Cross-Sectional Study Based on National Data"

_healthcare, 2025, doi:10.3390/healthcare13202651_

Round 1

Reviewer 1 Report

Comments and Suggestions for Authors

This article presents a significant and timely analysis of radiological workforce in Saudi Arabia. It is a strong study because it uses national registry data from the SCFHS, making it a trustworthy source of evidence and an important addition to the medical literature, especially given the implications of healthcare policy and planning under Vision 2030. And there are several areas of concern. 
Most notably, there is lack of relationship with workload or technology. In some cases, imaging equipment availability (CT, MRI, PET-CT scanners), and even the volumes of examinations, were omitted from the analysis. The number of radiologists is only one aspect; their capacity is based on their workloads and the technology available.  A radiologist working in a rural practice with one old CT scanner has a very different capacity than another working in an urban center with five different, state-of-the-art modalities. Clinical work-load and technology use should be considered in future studies.

There is a very superficial discussion of Teleradiology. It is highlighted as a 'possible answer to geographic disparities', yet the analysis from both a technical perspective and implementation perspective could be addressed to a greater extent. What is the state of teleradiology infrastructure in SA? What medical and data security barriers exist? How can we use it specifically to leverage subspecialty schooling (eg a central pediatric neuroradiologist who will read cases from various regions)? The "Young Workforce" makes for a much more challenging discussion, even though the youth of the workforce is framed as a primary strength. However, to consider it critically, we must probe the implications to the negative: a limited number of senior mentors, training burden with a lot of registrars and senior registrars needing to be developed to consultancy level, and that will take a lot of training. 

"Youthfulness in the workforce" is a double-edged sword. Youthfulness in the workforce is discussed mainly as a strength. A critique would look to highlight some potential shortcomings: a lack of experience in senior mentors and the significant burden of education to bring this large cohort of registrars and senior registrars to a consultancy level, which takes considerable educational resources.
The category of "overseas" is somewhat vague. It would have been more informative to know which countries are the main sources of foreign-trained radiologists (e.g., other Arab countries, North America, Europe, Asia) when much of the specificity of training, and quality of training, varies widely.

Author Response

I would like to thank the reviewer for their thoughtful and constructive feedback on the manuscript. Their comments have been invaluable in enhancing the clarity, depth, and overall relevance of the study

Reviewer 1

Comments 1: This article presents a significant and timely analysis of radiological workforce in Saudi Arabia. It is a strong study because it uses national registry data from the SCFHS, making it a trustworthy source of evidence and an important addition to the medical literature, especially given the implications of healthcare policy and planning under Vision 2030. And there are several areas of concern. 
Most notably, there is lack of relationship with workload or technology. In some cases, imaging equipment availability (CT, MRI, PET-CT scanners), and even the volumes of examinations, were omitted from the analysis. The number of radiologists is only one aspect; their capacity is based on their workloads and the technology available.  A radiologist working in a rural practice with one old CT scanner has a very different capacity than another working in an urban center with five different, state-of-the-art modalities. Clinical work-load and technology use should be considered in future studies.

Response 1: Thank you for pointing this out. I agree that workforce capacity is closely linked to workload and technology availability. Accordingly, I have added this limitation in the discussion and emphasized that future studies should integrate clinical workload and the availability and distribution of advanced imaging modalities when assessing radiologist capacity.

Comments 2: There is a very superficial discussion of Teleradiology. It is highlighted as a 'possible answer to geographic disparities', yet the analysis from both a technical perspective and implementation perspective could be addressed to a greater extent. What is the state of teleradiology infrastructure in SA? What medical and data security barriers exist? How can we use it specifically to leverage subspecialty schooling (eg a central pediatric neuroradiologist who will read cases from various regions)? The "Young Workforce" makes for a much more challenging discussion, even though the youth of the workforce is framed as a primary strength. However, to consider it critically, we must probe the implications to the negative: a limited number of senior mentors, training burden with a lot of registrars and senior registrars needing to be developed to consultancy level, and that will take a lot of training. 

Response 2: Thank you for this valuable comment. I have expanded the discussion on tele radiology in Saudi Arabia including the launch of the national platform of teleradiogy by the Ministry of health including its benefits. I also emphasized its role in supporting subspecialty reporting and training. In addition, I revised the section on the young workforce to highlight not only its strengths in sustainability but also challenges related to mentorship gaps and the training burden required to advance registrars to consultancy level.

Comments 3: "Youthfulness in the workforce" is a double-edged sword. Youthfulness in the workforce is discussed mainly as a strength. A critique would look to highlight some potential shortcomings: a lack of experience in senior mentors and the significant burden of education to bring this large cohort of registrars and senior registrars to a consultancy level, which takes considerable educational resources.
The category of "overseas" is somewhat vague. It would have been more informative to know which countries are the main sources of foreign-trained radiologists (e.g., other Arab countries, North America, Europe, Asia) when much of the specificity of training, and quality of training, varies widely.

Response 4: Thank you for this important comment. I have revised the discussion of the “youthful workforce” and acknowledge the challenges

Reviewer 2 Report

Comments and Suggestions for Authors
  1. Introduction page 1 line 39. The sentence “Radiology plays a crucial role in contemporary healthcare by supporting early diagnosis, treatment planning, and follow-up across medical disciplines” should be further supported by citing https://pmc.ncbi.nlm.nih.gov/articles/PMC10919692/
  2. Method page 3 line 100. Instead of only stating ‘various subspecialties’, please specify which subspecialties were included.
  3. Clearly state the inclusion and exclusion criteria in the methods section.
  4. Discussion page 7 line 249. Add citation of https://doi.org/10.52225/narrax.v2i3.180 to the sentence “incorporating telemedicine competencies into training can help prepare the workforce for the hybrid service models”.

Author Response

Comments 1: Introduction page 1 line 39. The sentence “Radiology plays a crucial role in contemporary healthcare by supporting early diagnosis, treatment planning, and follow-up across medical disciplines” should be further supported by citing https://pmc.ncbi.nlm.nih.gov/articles/PMC10919692/

Response 1: Thank you for this valuable comment. The sentence has been updated and is now supported by the recommended reference, which has been cited in the Introduction “Marlina YS, Novirianthy R, Beočanin A. Palliative radiotherapy for leptomeningeal metastases after photon-based intensity-modulated radiotherapy in a nasopharyngeal cancer patient. Narra J. 2023 Dec;3(3):e266. doi: 10.52225/narra.v3i3.266. Epub 2023 Oct 12. PMID: 38455636; PMCID: PMC10919692”

Comments 2: Method page 3 line 100. Instead of only stating ‘various subspecialties’, please specify which subspecialties were included.

Response 2: Thank you for this supportive suggestion. I have revised the Methods section to clearly list the subspecialties instead of using the general phrase “various subspecialties.

Comments 3: Clearly state the inclusion and exclusion criteria in the methods section.

Response 3: Thank you for this important comment. I have revised the Methods section and clearly state the inclusion and exclusion criteria used in this study.

Comments 4: Discussion page 7 line 249. Add citation of https://doi.org/10.52225/narrax.v2i3.180 to the sentence “incorporating telemedicine competencies into training can help prepare the workforce for the hybrid service models”.

Response 4: Thank you for this valuable comment. The discussion has been updated and is now supported by two references. “Okafor, C. (2024). Enhancing primary healthcare delivery in Nigeria through the adoption of advanced technologies. Narra X, 2(3). https://doi.org/10.52225/narrax.v2i3.180”. “Porterfield L, Warren V, Schick V, Gulliot-Wright S, Temple JR, Vaughan EM. Addressing Training Gaps: A Competency-Based, Telehealth Training Initiative for Community Health Workers. Telemed Rep. 2023 Jun 16;4(1):126-134. doi: 10.1089/tmr.2023.0007. PMID: 37351464; PMCID: PMC10282968”.

Reviewer 3 Report

Comments and Suggestions for Authors

Radiology is considered a medical shortage profession worldwide, and therefore an analysis of the supply of radiologists in a high-income Middle Eastern country, their distribution by age and subspecialty, may be of interest. The draft is based on available data, is descriptive, and international comparisons are valuable.
However, the relative shortage and the reasons for the unequal distribution could have been clarified through a short questionnaire survey and personal interviews, and a comparison with the surrounding richer Gulf countries (UAE, etc.) would be interesting. Radiology is increasingly a telemedical profession, but the very low number of interventional radiologists in Saudi Arabia is worrying. It is also worth finding out about the motivation, equipment, and financing background in this regard.

Author Response

Comments 1: Radiology is considered a medical shortage profession worldwide, and therefore an analysis of the supply of radiologists in a high-income Middle Eastern country, their distribution by age and subspecialty, may be of interest. The draft is based on available data, is descriptive, and international comparisons are valuable. However, the relative shortage and the reasons for the unequal distribution could have been clarified through a short questionnaire survey and personal interviews, and a comparison with the surrounding richer Gulf countries (UAE, etc.) would be interesting. Radiology is increasingly a telemedical profession, but the very low number of interventional radiologists in Saudi Arabia is worrying. It is also worth finding out about the motivation, equipment, and financing background in this regard.

Response 1: Thank you for this valuable comment. I acknowledge the global shortage of radiologists and note that our study was limited to registry data from the SCFHS, which did not capture institutional factors such as equipment or funding. To strengthen the manuscript, I have expanded the Discussion to compared to other Gulf Cooperation Council countries

Reviewer 4 Report

Comments and Suggestions for Authors

Abstract
- The stated objective in the abstract is not fully aligned with the objective presented in the Background section. Please ensure consistency.
- The keyword “Saudi Arabia” is unnecessary, as it is already mentioned in the title.

Background
- Line 46: It seems excessive to cite six references (8–13) for a single sentence. Please reassess and retain only the most relevant sources.
- The research problem and the specific gap this study intends to address are not sufficiently clear.
- The significance and urgency of conducting this study need to be emphasized more strongly.

Methods
- Please provide a clear statement of the inclusion and exclusion criteria applied in this study. This will improve methodological transparency and reproducibility.

Results
- Avoid repeating all numerical values in the text; highlight only the essential findings and direct readers to the corresponding tables or figures.
- The section currently contains an excess of numerical detail. Instead, clearly emphasize the key messages and how the results relate to the study’s objectives.

Discussion
The discussion would benefit from a more critical perspective. Specifically, please elaborate on:
- The underlying reasons for workforce disparities,
- The potential impact of subspecialty shortages on healthcare service delivery, and 
- The most urgent policy actions that should be prioritized.

References
- The references are not consistently formatted. Issues include repeated numbering, inconsistent presentation of DOIs, and irregularities in journal name formatting. Please revise the entire reference list to comply with the journal’s required style.

Author Response

Abstract

Comments 1: - The stated objective in the abstract is not fully aligned with the objective presented in the Background section. Please ensure consistency.

Response 1: Thank you for this helpful comment. I have revised the abstract objective to ensure consistency with the Background section.

Comments 2: - The keyword “Saudi Arabia” is unnecessary, as it is already mentioned in the title.

Response 2: Thank you for the observation. The keyword “Saudi Arabia” has been removed.

Background

Comments 3: - Line 46: It seems excessive to cite six references (8–13) for a single sentence. Please reassess and retain only the most relevant sources.

Response 3: Thank you for this comment. The citations have been updated

Comments 4: - The research problem and the specific gap this study intends to address are not sufficiently clear.

Response 4: Thank you for this important comment. I have revised the Introduction to more clearly state the research problem and to highlight the specific knowledge gap.

Comments 5: - The significance and urgency of conducting this study need to be emphasized more strongly.

Response 5: Thank you for this valuable comment. I have revised the Introduction to emphasize the significance and urgency of this study.

Methods

Comments 6: - Please provide a clear statement of the inclusion and exclusion criteria applied in this study. This will improve methodological transparency and reproducibility.

Response 6: Thank you for this helpful comment. I have revised the Methods section and provided a clear statement of the inclusion and exclusion criteria.

Results

Comments 7: - Avoid repeating all numerical values in the text; highlight only the essential findings and direct readers to the corresponding tables or figures. The section currently contains an excess of numerical detail. Instead, clearly emphasize the key messages and how the results relate to the study’s objectives.

Response 7: Thank you for this comment. I have revised the Results section to avoid repetition of numerical values.

Discussion

Comments 9: The discussion would benefit from a more critical perspective. Specifically, please elaborate on:

- The underlying reasons for workforce disparities,

- The potential impact of subspecialty shortages on healthcare service delivery, and

- The most urgent policy actions that should be prioritized.

Response 9: Thank you for this constructive comment. I have revised the Discussion to provide a more critical perspective.

References

Comments 10: - The references are not consistently formatted. Issues include repeated numbering, inconsistent presentation of DOIs, and irregularities in journal name formatting. Please revise the entire reference list to comply with the journal’s required style.

Response 10: I appreciate the reviewer’s observation regarding inconsistencies in the reference list. I have carefully revised the entire reference section to ensure consistency with the journal’s required style.

Reviewer 5 Report

Comments and Suggestions for Authors

The manuscript addresses a relevant and underexplored topic.

  1. Introduction

    • The introduction provides adequate context but could be improved by a more critical synthesis of previous regional studies (e.g., comparisons with Gulf Cooperation Council countries) and clearer justification of why this study fills a knowledge gap.

  2. Methods

    • The design is appropriate for the objectives. However, the methods are described only at a descriptive level. Please clarify how missing or “unknown” data (7% of radiologists) were handled.

    • Consider adding inferential analyses (e.g., associations between professional rank and geographic region, or between age and subspecialty). This would strengthen the scientific contribution.

  3. Results

    • The finding that two-thirds of radiologists trained abroad deserves further emphasis in results and a clearer link to policy implications.
  4. Discussion

    • While the discussion compares Saudi Arabia with international benchmarks, it should be more critical regarding workforce sustainability, subspecialty shortages, and rural–urban disparities.

    • More concrete recommendations for policy and training (e.g., fellowship expansion, incentives for rural practice, integration of teleradiology) would increase the impact of the study.

    • The limitations section is too brief. It should include: (a) reliance on self-reported subspecialty classification, (b) lack of workload/equipment data, (c) cross-sectional design.

Author Response

Comments 1: The manuscript addresses a relevant and underexplored topic.

Introduction

Response 1: Thank you for this positive comment. I appreciate this acknowledgment.

Comments 2: The introduction provides adequate context but could be improved by a more critical synthesis of previous regional studies (e.g., comparisons with Gulf Cooperation Council countries) and clearer justification of why this study fills a knowledge gap.

Response 2: Thank you for this helpful comment. I have included a more critical synthesis of previous regional studies, with comparisons to other Gulf Cooperation Council countries. I have also clarified the knowledge gap.

Methods

Comments 3: The design is appropriate for the objectives. However, the methods are described only at a descriptive level. Please clarify how missing or “unknown” data (7% of radiologists) were handled.

Response 3: Thank you for this helpful comment. I have revised the Methods section to clarify how missing or “unknown” data were handled.

Comments 4: Consider adding inferential analyses (e.g., associations between professional rank and geographic region, or between age and subspecialty). This would strengthen the scientific contribution.

Response 4: Thank you for this valuable suggestion. I have expanded the analysis by examining the associations between professional rank and work region. Both analyses revealed statistically significant associations, with results presented in the new tables and described in the Results.

Results

Comments 5: The finding that two-thirds of radiologists trained abroad deserves further emphasis in results and a clearer link to policy implications.

Response 5: Thank you for this insightful comment. I have added a statement in the Discussion highlighting the policy implications of this reliance on foreign-trained radiologists, particularly in terms of workforce sustainability, quality assurance, and the need to expand local training capacity.

Discussion

Comments 6: While the discussion compares Saudi Arabia with international benchmarks, it should be more critical regarding workforce sustainability, subspecialty shortages, and rural–urban disparities.

Response 6: Thank you for this constructive comment. I have expanded the Discussion

Comments 7: More concrete recommendations for policy and training (e.g., fellowship expansion, incentives for rural practice, integration of teleradiology) would increase the impact of the study.

Response 7: Thank you for this valuable suggestion. We have revised the Discussion to include more concrete policy and training recommendations.

Comments 8: The limitations section is too brief. It should include: (a) reliance on self-reported subspecialty classification, (b) lack of workload/equipment data, (c) cross-sectional design.

Response 8: Thank you for this helpful comment. We have expanded the Limitations section

Round 2

Reviewer 1 Report

Comments and Suggestions for Authors

Although the author references the national teleradiology platform, no indication of the current penetration, use figures, or geographic spread is provided. Add a sentence with a reference to a current SA national study or report on the use of teleradiology adoption (e.g., Altuwaili et al., 2024, as cited).

The author did not split the "abroad" category into regions/nations, thus limiting the ability to view mobility pattern as well as quality of training. A note of acknowledgment of this limitation during the Discussion or a citation to future research to examine origin nations.

The statement states that there are deficits in mentorship but doesn't connect this to trackable effects (e.g., education quality, subspecialty development).  Insert a sentence such as: "This demographic shift can inhibit subspecialty development because of a lack of senior leadership and availability of mentorship."

The research still uses descriptive statistics only. There was a reference to a chi-square test in v2 (p. 7), but this was not completed within results or discussion. Discuss the importance of regional-rank associations in the Results or Discussion in brief.

Author Response

Comment 1: Although the author references the national teleradiology platform, no indication of the current penetration, use figures, or geographic spread is provided. Add a sentence with a reference to a current SA national study or report on the use of teleradiology adoption (e.g., Altuwaili et al., 2024, as cited).

Response 1: Thank you for this valuable and specific comment. I agree and I have added a sentence summarizing the most recent national findings on teleradiology implementation and cited Altuwaili et al. (2024).

Comment 2: The author did not split the "abroad" category into regions/nations, thus limiting the ability to view mobility pattern as well as quality of training. A note of acknowledgment of this limitation during the Discussion or a citation to future research to examine origin nations.

Response 2: Thank you for this insightful comment. I agree that identifying the countries or regions of foreign-trained radiologists would provide deeper insight into workforce mobility and training quality. However, the available registry data did not include details on the specific origin of foreign qualifications. I have now acknowledged this as a limitation

Comment 3: The statement states that there are deficits in mentorship but doesn't connect this to trackable effects (e.g., education quality, subspecialty development). Insert a sentence such as: "This demographic shift can inhibit subspecialty development because of a lack of senior leadership and availability of mentorship."

Response 3:
Thank you for this constructive suggestion. I have revised the relevant section of the Discussion

Comment 4: The research still uses descriptive statistics only. There was a reference to a chi-square test in v2 (p. 7), but this was not completed within results or discussion. Discuss the importance of regional-rank associations in the Results or Discussion in brief.

Response 4: Thank you for this important and constructive comment. I have added a statement in the section of Professional rank and work region and I acknowledge that inferential analysis is essential to enrich the interpretation of the data.

Reviewer 3 Report

Comments and Suggestions for Authors

Radiologist availability is a key issue in modern diagnostics, as the evaluation of increasingly sophisticated imaging procedures requires specialists, even beyond the application of AI. The draft has been supplemented and made more nuanced following the reviewer's advice. Therefore, it can be recommended for publication. 

Author Response

comment: Radiologist availability is a key issue in modern diagnostics, as the evaluation of increasingly sophisticated imaging procedures requires specialists, even beyond the application of AI. The draft has been supplemented and made more nuanced following the reviewer's advice. Therefore, it can be recommended for publication. 

Response: Thank you very much for this positive and encouraging comment. I greatly appreciate the reviewer’s recognition of the improvements made and the acknowledgment of the study’s contribution to addressing the critical issue of radiologist availability in modern diagnostics.